# Chest-Wall Tumors and Surgical Techniques: State-of-the-Art and Our Institutional Experience

**DOI:** 10.3390/jcm11195516

**Published:** 2022-09-20

**Authors:** Alessandro Gonfiotti, Alberto Salvicchi, Luca Voltolini

**Affiliations:** Thoracic Surgery Unit, Careggi University Hospital, 50134 Florence, Italy

**Keywords:** chest-wall tumors, chest-wall resection, chest-wall reconstruction

## Abstract

The chest wall can be involved in both primary and secondary tumors, and even today, their management and treatment continue to be a challenge for surgeons. Primary chest-wall tumors are relatively rare and include a large group of neoplasms that can arise from not only bone or cartilage of the chest wall but also from associated subcutaneous tissue from muscle and blood vessels. Secondary tumors refer to a direct invasion of the chest wall by neoplasms located elsewhere in the body, mainly metastases from breast cancer and lung cancer. En-bloc surgical excision of the lesion should ensure adequate negative margins to avoid local recurrence, and a full thickness surgical resection is often required, and it can result in important chest-wall defects such as skeletal instability or impaired breathing dynamics. The reconstruction of large defects of the chest wall can be complex and often requires the use of prosthetic and biologic mesh materials. This article aims to review the literature on these tumor entities, focusing on the main surgical techniques and the most recent advances in chest-wall resection and reconstruction. We also report on the institutional experience our center.

## 1. Introduction

The chest wall, or thoracic wall, represents a complex musculoskeletal that ensures structural integrity, offers stability for movement of the shoulder and upper arms, and provides protection to the vital organs contained within. The thoracic wall is also involved in normal respiratory function.

The chest wall can be home to both benign and malignant tumors.

Chest-wall tumors are heterogeneous group of lesions that can be subdivided into primary and secondary tumors. Primary tumors originate from anatomical structures that make up the rib cage, such as muscles, soft tissue, blood vessels, nerves, cartilage, or bone.

Secondary tumors of the chest wall can arise from direct metastasization by breast and lung cancer or from neoplasm located elsewhere in the body, and their incidence is higher than that of primary tumors.

Approximately 50% of all chest-wall tumors are malignant in nature [1].

The management of chest-wall tumors is still a major diagnostic and therapeutic challenge for thoracic surgeons, but the considerable progress achieved in recent years have led to an improvement in the prognosis and long-term survival thanks also to an advance of surgical techniques and prosthetic materials adopted for reconstruction.

Surgical resection is often the best treatment strategy and offers the best chance of cure, and the involvement of ribs, sternum, and spine should not be considered an absolute contraindication to the surgical approach.

En-bloc surgical excision of the lesion should ensure adequate margins of healthy tissue around the tumor mass to avoid local recurrence.

In some cases, after surgical resection, the chest-wall reconstruction is necessary, with the aim of restoring anatomical defects and reducing skeletal instability, paradoxical respiratory motion, respiratory failure, and infection diseases [2].

A multidisciplinary approach involving more than one professional, including thoracic and plastic surgeons, oncologists, radiotherapists, anesthesiologists, and physiotherapists, is essential in the face of advanced diseases requiring demolition and complex reconstruction.

The management of chest-wall tumors is not always easy and still represents one of the most important and difficult challenges for surgeons and oncologists.

This article aims to review the literature on these tumor entities, focusing on the main surgical techniques and the most recent advances in the chest-wall resection and reconstruction.

## 2. Primary Chest-Wall Tumors

Primary chest-wall tumors are infrequent, and they represent a wide and heterogeneous group of neoplasms that includes both benign and malignant neoplasms

Primary tumors of the chest wall are rare and count less than 2% of all cancers occurring in the chest, and about 50% to 80% of these are malignant. Generally, the classification of primary tumors is based on their proposed tissue of origin (soft tissue or bone): 55% originates from bone or cartilage, and the remaining 45% arises from soft tissue [3].

Approximately 85% of all primary bone tumors of the chest wall originate from the ribs, and 88% of these are considered malignant. The remaining 15% affects the sternum and the sternal tumors are almost always malignant [4]. More than 20% of primary cancers are incidental findings on routine radiography carried out for other reasons.

### 2.1. Primary Osseous Tumors

Usually, malignant bone tumors of the chest wall do not have a good prognosis, showing an average 5-year survival of about 60% [5].

Unlike benign lesions, malignant lesions tend to grow faster, and the most typical clinical presentation is pain with or without a palpable mass, and the painful condition is often misinterpreted as chronic musculoskeletal pain or as the result of chest trauma. In about 20% of patients, bone cancer is discovered incidentally by radiography of the chest for other reasons.

However, these tumors can have a largely variable clinical presentation, and it is often difficult to distinguish malignant forms from benign ones only by the symptoms: for example, in Ewing’s sarcoma or Langerhans Cell Histiocytosis, non-specific symptoms, such as generalized malaise, including bone pain, fever, and asthenia, may occur [6].

The suspicion of malignant tumor should arise from the detection of any mass in a child’s chest wall or a sternal mass in the adult. A fixed hard mass that is tightly attached to the chest wall increases the likelihood of a bone tumor [4].

Therefore, a rigorous history evaluation is crucial in the diagnostic process of these tumors. Patient assessment should include a full anamnesis of prior neoplastic pathologies and previous exposure to radiation or chest trauma, and a careful physical examination is important, because it allows us to confirm the presence of the tumor, its size, and its location; it also and helps to establish the next diagnostic/surgical planning.

The most common primary bone tumors of the chest wall are shown in Table 1.

### 2.2. Primary Soft-Tissue Tumors

Soft-tissue tumors account for about 45% of all chest-wall malignancies. Soft-tissue sarcomas are the most frequent malignant tumors of the chest wall and typically arise in middle-aged men, with the exception of rhabdomyosarcoma, which is more frequent in pediatric age. A list of the main primary soft-tissue tumors is given in Table 2.

Primary soft-tissue tumors often appear as a palpable and indolent mass at the chest wall.

Patients with pain and rapid growth of tumor mass are typically affected by a malignant-behavior tumor, with likely invasion of surrounding structures. However, there are no clinical features that can help distinguish malignant from benign cancer. Initial diagnosis processing consists of obtaining a detailed clinical history, including the search for previous neoplasms, radiation exposure, and the symptomatology associated with mass [3].

## 3. Metastatic Lesion of Chest Wall

The chest wall may also be affected by metastatic lesions by direct invasion of adjacent anatomical structures or by haematogenic and lymphatic diffusion. Secondary thoracic wall tumors have a higher incidence than primary tumors, and the most common are breast and lung cancer metastases. In the case of secondary tumors, the chest wall and sternal resection could have a palliative role, with the aim of controlling the pain the ulceration, as well as possible bleeding and infections caused by the tumor mass. In surgical resection and possible reconstruction of the chest wall, surgeons must consider previous therapies and surgeries performed for the treatment of primitive tumors [1].

## 4. Diagnosis: Imaging and Tissue Biopsy

### 4.1. Imaging

A radiological diagnosis of chest-wall tumors can be a challenge for physicians due to considerable histological heterogeneity. In addition, diagnostic imaging cannot clearly distinguish between benign and malignant tumors [7].

In a radiological evaluation, it is essential to integrate the various methods available: the most used and recommended for the evaluation of chest-wall tumors are chest X-rays, computed tomography (CT), magnetic resonance imaging (MRI), and positron emission tomography (PET).

Chest X-rays are commonly the initial radiological method. The chest X-ray allows us to locate the lesion and its size, as well as to detect the presence of calcification, erosion, and bone destruction. Many bone tumors of the chest wall have a peculiar pathognomonic characteristic and can be identified with a chest X-ray, but early stage tumors could be lost. Sometimes conventional radiography is enough to diagnose without the need to perform a tissue biopsy [4].

CT is a useful radiological investigation because it is more sensitive and specific than chest X-rays and offers better image resolution.

CT, especially if contrast medium is used, allows us to evaluate the extent of the tumor and the involvement of adjacent structures, such as lung, pleura, mediastinum, and lymph node invasion. It also provides useful information on the vascularity composition and density of the cancer lesion.

In the diagnostic process, MRI plays a crucial role in providing accurate tissue characterization and superior spatial resolution. In addition, MRI facilitates the differentiation of chest-wall tumors from infectious or inflammatory processes [8].

PET is a non-invasive radiological investigation that allows us to evaluate the presence and extension of the disease. PET is particularly advantageous in staging disease and in assessing treatment response, especially in sarcomas. Schwarzbach et al. (2000) [9] reported 88% sensitivity and 92% specificity of PET in detecting high-grade local sarcoma recurrence.

### 4.2. Tissue Biopsy

In most cases, radiological investigations are insufficient to reach the definitive diagnosis, and often histological confirmation is needed; therefore, tissue biopsy is used to confirm diagnosis. Usually for lesions over 2 cm, it is preferable to have a preoperative diagnostic confirmation.

Pathological tissue samples can be obtained through fine-needle aspiration, incisional biopsy, or excisional biopsy. When metastatic lesions are suspected, it is reasonable to use fine-needle aspiration; in addition, several authors reported that core needle biopsy in bone tumors has an accuracy comparable to that of incisional biopsy [10]. However, the small amount of tissue obtained with fine-needle aspiration is often insufficient in order to achieve a definitive diagnosis. When the tumor has a size greater than 5 cm, an incisional biopsy can be performed, keeping in mind that, in the case of definitive surgical resection, re-excision of the biopsy site is necessary. An excisional biopsy can be planned in the case of a small lesion (<2 cm), and wide negative margins must be guaranteed (at least 2 cm). Excisional biopsy has both a diagnostic and therapeutic purpose [4].

## 5. Surgical Management

The management of the chest-wall tumor is based on the site, size, histology, and stage of the tumor. In addition, the patient’s age and the presence of comorbidity must be taken into account.

In this scenario, surgical resection of chest-wall tumors with possible prosthetic reconstruction is a primary treatment. In the case of chest-wall metastases, surgical resection can prolong the overall survival and improve the quality of life. However, not all chest-wall tumors are susceptible to primary surgical excision and neoadjuvant chemotherapy; radiotherapy may also be required. Several tumors are highly chemo-sensitive, and induction therapy is critical to reducing tumor mass [1]. Among chest-wall tumors, osteosarcoma, Ewing sarcoma, and rhabdomyosarcoma should be treated with preoperative chemotherapy, and on the basis of the therapeutic response, the surgical option may be considered. Other chest-wall tumors, such as solitary plasmacytoma and Langerhans Cell Histiocytosis, require medical treatment, and surgery is generally not necessary.

### 5.1. Surgical Resection

The main intention of the surgery is to ensure an adequate oncology resection and to obtain appropriate disease-free margins. Positive or inadequate resection margins and tumor histotype are directly related to disease-free survival. A Mayo Clinic study involving 90 patients (71 pts had a malignant tumor) showed that patients with a 4 cm resection margin had a 5-year survival rate of 56% compared to 29% for patients with a 2 cm resection margin [11].

Although in the case of metastases, low-grade tumors or benign lesions may be sufficient 2 cm of margin, in the presence of malignant tumors, it is necessary to perform an excision that guarantees at least a margin of 4 cm remaining free from disease. For example, in the case of inadequate margins, chondrosarcoma has a recurrence rate of 73%, and sarcoma has a rate of 96% [12,13].

A separate case is the desmoid tumor; although it is considered to be a benign soft-tissue tumor of the chest wall, it is locally invasive and has a high rate of recurrence, Abbas et al. [14]. reported a local relapse of 89% with positive margins versus 18% with negative margins; therefore, a clear margin at least of 5 cm should be warranted

In addition, for tumors with a high grade of malignancy, in order to achieve an oncological radicality, a surgical resection at full thickness is required, including muscle, bone, and possibly skin [15]. For aggressive tumor lesions that have a tendency to spread along the periosteum, it is mandatory to remove the entire bone (rib or sternum) without compromising respiratory function. Moreover, due to the increased risk of metastasis to the sub periosteum and adjacent structures, resection of the rib above and below the tumor is generally recommended [1,2,3,4]. The tumor should be removed “en bloc” with the possible scar of a previous biopsy, together with any other tissue or structure involved (soft tissue, pleura, lung, and diaphragm). In some cases, it may be useful to use video-assisted thoracic surgery (VATS) to evaluate the extent of the cancer and the involvement of the lung or other mediastinal organs. The thoracoscope should be positioned away from the lesion, owing to the risk of the tumor spreading. In chest-wall tumors, especially those with a high grade of malignancy, removal must be radical, and the extension of the oncological resection should not be compromised by fear of the chest defect and the concern of the subsequent reconstruction. Lindford et al. [16] showed that, in selected cases of extensive chest-wall recurrent breast cancer, wide resection is feasible and safe.

In our surgical policy of the chest-wall tumors, we perform a skin incision including the site of previous biopsy, tissues affected by cancer, and also tissues previously subjected to radiotherapy. We carry out a wide surgical excision involving tumor ribs with at least a resection margin of 3 cm. We also remove the rib above and below the tumor to ensure adequate resection margins [17]. In our center, the sternal tumors are treated with partial, subtotal, or total sternectomy depending on the lesion dimensions, but in all cases, we remove adjacent sternocostal cartilages bilaterally [18].

Moreover, we adopted Enneking classification, adapted to chest-wall surgery that classifies the surgical resection margins in wide (free margins > 2 cm from the tumor, R0), marginal (free margins but <2 cm, R0), or intralesional (microscopic or macroscopic infiltration of the margins) [19].

### 5.2. Surgical Reconstruction

The proper oncological resection of chest-wall tumors may require reconstruction to restore the structural and functional integrity of the thorax in order to ensure protection of the underlying organs; prevent lung herniation and paradoxical movements of the chest wall; and, where possible, provide acceptable aesthetic results. Currently there are no guidelines or consensus that fully describe the indications for chest-wall reconstruction; therefore, there are several surgical procedures that often depend on the knowledge and preferences of the surgeon [20].

It is generally accepted that not all chest-wall defects need to be repaired. Usually, small defects (<5 cm) or resections involving less than three ribs do not require reconstructive procedures, and the soft tissue alone is broadly sufficient to cover the chest-wall breach [21]. Subscapular and apico-posterior chest-wall defects up to 10 cm in size may not be reconstructed because scapula and shoulder ensure adequate support and stiffness [4,20].

Many surgeons agree that the indications for reconstruction after resection are as follows:
-Chest-wall defects larger than 5 cm in diameter or total area > 100 cm^2^;-Removal > 3 ribs from the anterior chest wall;-Removing > 4 ribs from the posterior chest wall;-In the case of posterior resections (included small defects), below the fourth rib should be reconstructed to avoid scapular entrapment [22].

Chest-wall reconstruction depends on the extent of the resection, partial- or full-thickness defects. In the case of full-thickness resections, the reconstructive time should be performed in the same surgery in order to reestablish the structural integrity of the skeletal, protect vital organs, and avoid impairing respiratory function. The fear of reconstruction should not limit the extent of the resection; therefore, it is important to carefully plan the surgery involving other professionals (e.g., plastic surgeon), and if the surgeon’s expertise is not suited to the task, the patient should be directed to experienced centers [4].

There are multiple options available for chest-wall reconstruction and stabilization, including soft-tissue coverage and prosthetic materials. Currently, surgeons have different types of prosthetic material to restore chest the wall, such as synthetic materials, alloplastic, and biologic materials [21].

The suitable and ideal material for chest-wall prosthetic reconstruction should be rigid enough to prevent paradoxical movements, quite malleable, strongly resistant to infection, biologically inert, radiolucent, and cheap [20].

However, there is still no ideal material, and often these materials are used in combination. Each prosthetic material for reconstruction has advantages and disadvantages, and none has demonstrated a clear superiority over the others; moreover, the method of reconstruction often depends on the surgeon’s preference and skills. [23,24,25].

Azoury et al. (2016) [26] created a schematic technical guide that can help surgeons in chest-wall reconstruction: in the case of small defects (<5 cm or <2 ribs resected), it is enough to use muscle flap or soft-tissue coverage alone. If the portion to be repaired is greater (>5 cm; >2 resected ribs or >10 cm back), you can use synthetic mesh with soft tissue or muscle flap; however, if there is a high risk of contamination and complications, the authors advise to use a biologic inlay with a synthetic onlay and muscle flap.

#### 5.2.1. Synthetic Materials

The surgeon has many synthetic materials available for the reconstruction which can be distinguished as flexible (e.g., PTFE) and rigid (e.g., Methyl-methacrylate) (Table 3). Vicryl and polypropylene meshes are flexible and malleable; therefore, they can be evenly elongated in all directions. This allows an even distribution of the tension along the edges of the chest-wall defect. These meshes are permeable, avoiding seromas. Polytetrafluoroethylene (PTFE) is a similar flexible prosthetic material that can be used to obtain a watertight closure. Large chest-wall defects can be repaired with PTFE, but soft-tissue coverage or muscle flap are needed. PTFE use is absolutely contraindicated in case of infection [20,23]. Methyl methacrylate is another prosthetic material that can be used to restore the rigidity of the chest wall after extensive rib or sternal resections. Methyl methacrylate is a resin that is commonly used in association with polypropylene in a “sandwich technique”: A first layer of polypropylene meshes is placed to cover the defect, and then a methyl methacrylate substitute is added. Finally, a second layer of polypropylene is put to cover the methyl methacrylate, and through an exothermic reaction, the resin hardens. Some authors argued that the rigidity of methyl methacrylate favors the onset of pain and atelectasis; also, not being permeable facilitates seromas and wound infection [20]. Weyant et al. [24] reported that the wound infection rate at 90 days was 10–20%, and 5% of patients required the removal of the prosthesis.

#### 5.2.2. Bioprosthetic Materials

In the current setting, the bioprosthetic materials represent an evolution in chest-wall reconstruction.

Over the last 20 years, several biological meshes have been designed from human (allograft, e.g., dermis, intestinal mucosa, or pericardium) or animal (xenograft; porcine or bovine) tissues (Table 4). The decellularized biological meshes’ function is to create a solid scaffold for growth and healing tissue through a gradual process of revascularization and remodeling by autologous tissues. It is able to stimulate the differentiation of mesenchymal stem cells in the bone marrow and increase fibroblasts’ proliferation by 2/3 times. Another important feature is their resistance to infections [21], and some authors suggest that, in case of infections, they should not be removed [27,28].

Giordano et al. (2020) carried out the first comparative study between the acellular dermal matrix and synthetic meshes. They retrospectively evaluated a cohort of 146 patients who underwent a chest-wall reconstruction, using an acellular dermal matrix or synthetic meshes; the primary endpoint was surgical-site complications. In this study, the authors reported that the surgical-site-complications rate was higher in the synthetic-meshes group than in the acellular-dermal-matrix prostheses group (32.6% vs. 15.7%; *p* = 0.027). In addition, in many cases of infection, it was not necessary to remove bioprosthetic meshes [29].

The authors prefer to use biological prosthetic material (porcine dermis, e.g., Permacol) in chest-wall reconstruction, because it is well incorporated into the surrounding tissues, easy and quick put in place, and is easily fixed under tension to the edges defect (Figure 1). Finally, the biological meshes demonstrated a significant limitation to air and fluid and a high resistance to infections.

Synthetic materials can give rise to various complications, including infections, as they represent a foreign body; in our experience, we had to perform several redo-surgeries, with removal of the infected synthetic meshes. In the case of large anterior chest-wall defects, to give strength to the reconstruction, we often use biological prosthesis in combination with rigid systems (e.g., titanium bar) and a myocutaneous flap (Figure 2).

The reconstruction of the chest wall is a real challenge, we believe that biological materials are a valid alternative, especially in the case of infected fields or in patients at a high risk of infection. Moreover, the biological meshes favor a good wound healing and a long-term stability, with few complications in the post-operative period [18], and they are also safe for pediatric patients.

#### 5.2.3. Osteosynthesis System

Osteosynthesis systems are intended to restore the structural integrity of the chest wall, avoiding paradoxical inward movements; to prolong mechanical ventilation; and to reduce chest-wall deformities with better cosmetic results. They can be used for stabilization and fixation after ribs or sternum resection: rib-to-sternum fixation; rib-to-rib fixation by plating transverse across the sternum. Actually, the most common osteosynthesis systems are STRATOS and MatrixRIB fixation (Table 5). The osteosynthesis system are titanium-based, a prosthetic material highly biocompatible, inert, and MRI compatible. Generally, they are used in combination with biological or synthetic meshes, with or without myocutaneous flaps [21].

According to many authors, the titanium system is the best choice to establish large full-thickness defects due to few complications (infections, bar fracture, or dislocation) [30,31].

In our experience we reported better results with the use of the STRATOS titanium rib bridge system, because it is simple to handle and to fix and allows for a variety of options for thoracic reconstruction. In the case of large and full-thickness defects, biological meshes are placed in the inner edges of the defect and fixed onto a titanium bar, which is inserted on the ribs for transverse reconstruction (Figure 3). If necessary, we add a muscular flap to complete the reconstruction and to prevent infections [32,33].

## 6. Conclusions

Surgical management is the keystone in the treatment of chest-wall tumors, and R0 en-bloc resection with margins free is needed. The potential of oncologic radical surgery has been widely demonstrated, and medical and technological advances have improved survival [34]. Even an R1 resection, with only microscopic residual disease, would have an unfavorable prognosis for the patient [35].

Generally speaking, many chest-wall resections do not require any prosthetic reconstruction or muscle flaps coverage. However, in about one-third of cases, it is necessary to perform a prosthetic reconstruction to restore the integrity and stability of the rib cage resulting from surgical resection.

In our surgical procedures, we perform a skin incision including the site of the previous biopsy, invaded skin, and/or previously irradiated tissues; we have respected 3 cm free margin anteriorly and posteriorly to the tumor, and normal ribs located above and below of the lesion were always removed.

In the case of posterior tumors, it may be necessary to resect the vertebrae transverse process to ensure adequate disease-free margins.

Large defects of the thorax should be reconstructed because the chest-wall stabilization with prosthetic materials decreases the need for prolonged mechanical ventilation, prevents paradoxical chest-wall movements, and protects mediastinal organs (hearth, lungs, and vessels) from injury and infection.

In our policy, the fundamental rule is that defects smaller than 5 cm in any location and those up to 10 cm in size posteriorly do not require prosthetic reconstruction [32].

There are many materials available for chest-wall reconstruction, such as rigid and semi-rigid synthetic meshes (i.e., sandwich of MMM and PTFE), biological/biocompatible prostheses (i.e., acellular dermal matrix), and rigid biocompatible component (i.e., titanium bars/meshes).

Generally, the prosthetic material used depends on surgeon’s preference and experience; the material should then be calibrated on each patient so as to reconstruct the oncologic resection defect. However, in case of redo surgery or infected or irradiated sites, several authors recommend the use of bioprosthetic meshes or titanium-based materials.

Recent technological advances have led to the development of new materials for the reconstruction of chest-wall defects, such as cryopreserved homograft and allografts; in addition, with the increasing use of 3D printing technology, it is possible to design custom-made protheses. Preoperative 3D modeling can provide a quantitative template for reconstructing, facilitate understanding of individual patient anatomy, and improve surgical planning.

Recently, in our center, we performed a replacement sternectomy by utilizing a customized 3D-printed titanium implant in a patient with metastatic breast cancer to the chest wall.

After a preoperative discussion of the clinical case by a multidisciplinary board, we performed a total sternectomy with partial resection of the third, fourth, and fifth ribs, bilaterally. In the presence of total sternectomy, a rigid reconstruction is strictly required to preserve the rib-cage respiratory function [18]. Then we successfully implanted a 3D custom-made titanium prothesis, and a latissimus dorsi muscle flap was used to cover the sternal implant (Figure 4).

The new frontiers in chest-wall reconstruction seem to be to use 3D porous scaffolds that are capable of recreating the ensemble of cell-instructive physicochemical and structural signals inducing the formation and remodeling of bone tissue. However, the currently available devices fail to recreate the anatomical and functional characteristics of bone tissue, due to its structural complexity and mechanical properties. Kon et al. (2021) [36] produced a scaffold through an innovative nanotechnological process based on biomorphic transformation of rattan wood into 3D porous calcium phosphate ceramics with bone-like architecture. Kon et al., with their results, demonstrated the ability of calcium phosphate scaffolds to promote the regeneration of bone defects.

Today, chest-wall surgery still remains an important challenge for thoracic surgeons, despite continuous technological advancement and the development of new materials. The treatment of chest-wall tumors must include a careful preoperative analysis of the clinical case involving other specialist figures.

## Figures and Tables

**Figure 1 jcm-11-05516-f001:**
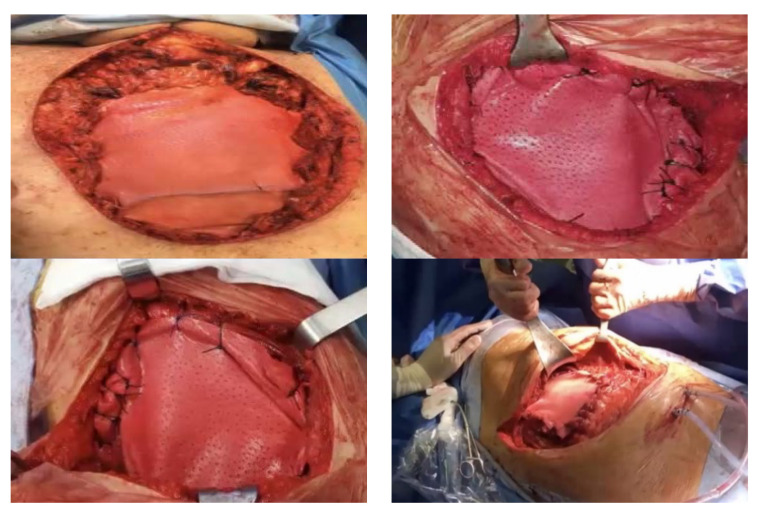
Acellular dermal matrix (Permacol; bovine dermis).

**Figure 2 jcm-11-05516-f002:**
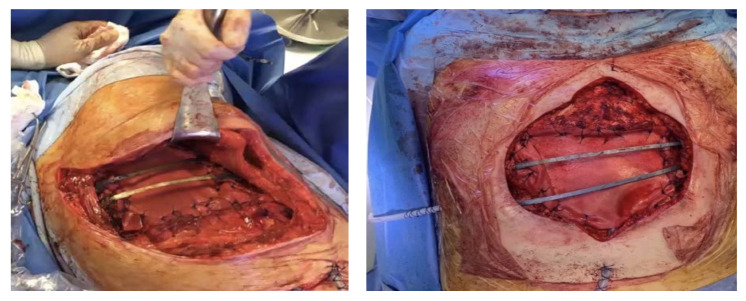
Acellular dermal matrix in combination with titanium rigid reconstruction.

**Figure 3 jcm-11-05516-f003:**
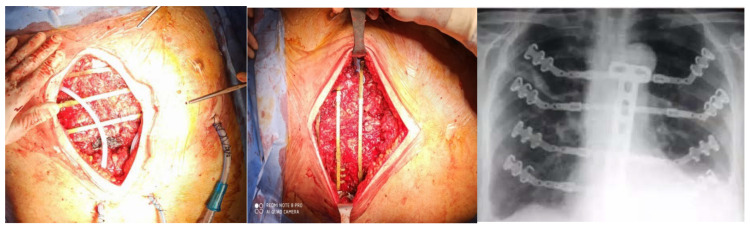
Rib bridge system with titanium-based bars (STRATOS).

**Figure 4 jcm-11-05516-f004:**
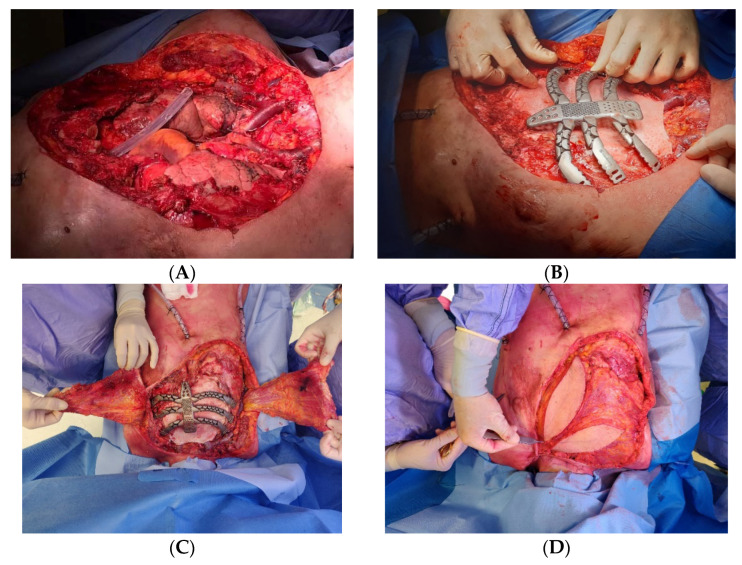
(**A**) Extended chest-wall resection for metastatic breast cancer. (**B**) Large defect coverage with biological prosthetic meshes and positioning of 3D custom-made titanium prothesis. (**C**,**D**) Latissimus dorsi myocutaneous flaps for sternal wound reconstruction.

**Table 1 jcm-11-05516-t001:** Classification of primary osseous chest-wall tumors.

**Benign Tumors**
Osteoblastoma, osteoid osteoma, chondroma, osteochondroma, benign chondroblastoma, fibrous dysplasia, eosinophilic granuloma, osteoclastoma, hemangioma, and aneurysmal bone cyst
**Malignant Tumors**
Osteosarcoma, chondrosarcoma, solitary plamacytoma, hemangiosarcoma, and Ewing sarcoma

**Table 2 jcm-11-05516-t002:** Classification of primary soft-tissue chest-wall tumors.

**Benign Tumors**
Hemangioma, glomus tumor, lymphangioma, schwannoma, neurofibroma, ganglioneuroma, paraganglioma, l–Lipoma, spindle cell lipoma, desmoid tumor, granuloma, and rhabdomyoma
**Malignant Tumors**
Leiomyosarcoma, rhabdomyosarcoma, angiosarcoma, malignant fibrous histiocytoma, aggressive fibromatosis, neuroblastoma, ganglioneuroblastoma, neurofibrosarcoma, liposarcoma, malignant lymphoma, and dermatofibrosarcoma protuberans

**Table 3 jcm-11-05516-t003:** Synthetic materials for the reconstruction.

Synthetic Materials
- Polyglactin (Vicryl) - Nylon - Polytetrafluoroethylene (Dualmesh) - Polypropylene (Marlex) - Methylmethacrylate - Silicone

**Table 4 jcm-11-05516-t004:** Biological meshes for the reconstruction.

Biological Meshes
- Porcine dermis (Permacol; XenMatrix; Strattice) - Porcine small intestine submucosa (Surgisis) - Bovine dermis (SurgiMend) - Bovine pericardium (Tutopatch; Veritas) - Cadaveric human dermis

**Table 5 jcm-11-05516-t005:** Osteosynthesis systems for the reconstruction.

Osteosynthesis Systems
- Titanium (Stratos; MatrixRIB Fixation) - Cadaveric bone - Stainless steel bars

## Data Availability

Not applicable.

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
