# Peer review of "Chest-Wall Tumors and Surgical Techniques: State-of-the-Art and Our Institutional Experience"

_jcm, 2022, doi:10.3390/jcm11195516_

Round 1

Reviewer 1 Report

The submitted manuscript Surgical Techniques in Chest Wall Tumors: Review Literature and Our Institutional Experience” is a non-systematic, narrative review of the available literature on the broad topic of chest wall tumors, their diagnosis and surgical management. In its extended introduction, the manuscript provides a general classification of primary and metastatic chest wall tumors and a general description of imaging and histological diagnostic methods for these tumors. The authors then focus on describing the state of the art in well-established surgical treatment.

The manuscript is clearly written and easy to read. However, its passive nature makes it a useful material for exam revision or extensive discussion than a scientific paper. The title of the manuscript suggesting surgical treatment of tumors does not reflect its content. The authors only mention the treatment methods they use without describing them scientifically.

I would recommend a thorough revision of this manuscript.

Author Response

Thanks to the reviewer for his work and we agree with his notes. The title of our manuscript may be misleading and we propose to change it to: “ CHEST WALL TUMORS AND SURGICAL TECHNIQUES: STATE OF THE ART AND OUR INSTITUTIONAL EXPERIENCE”.We wrote the manuscript based on the special issue proposed by the JCM: “Cardiothoracic Surgery: State of the Art and Future Perspectives”.Our manuscript aims to summarize the state of the art on chest wall tumors. It was not  our intention to build a systematic review.

Reviewer 2 Report

Dear Authors,

Thank you for your submission entitled "SURGICAL TECHNIQUES IN CHEST WALL TUMORS: REVIEW LITERATURE AND OUR INSTITUTIONAL EXPERIENCE".   Overall, the article provides a good review of past and present literature and experience in chest wall resection and also summaries the current preferences from the author's own institution.  I have a few comments/questions.

1) You mention in line 253 that "The suitable and ideal material for chest wall prosthetic reconstruction should be rigid enough to prevent paradoxical movements, quite malleable, strong resistance to infection, biologically inert, radiolucent, and cheap".   However, your institutional preference (line 307) includes the use of biologic mesh and titanium bars (when needed).   These materials, while they may fulfill some of your criteria, are also significantly more expensive that polypropylene mesh and/or methylmethacrylate. 

   a) Do you recommend biologic materials in ALL cases of chest wall reconstructions for defects >5cm, +3/4 ribs?   Are there situations where you still utilize synthetic materials? 

   b) You also mention the higher rates of infections in synthetic meshes.   Do you have institutional data that shows a significantly lower infection rate that can be added to this paper to support your use of these materials in all of your chest wall reconstructions?  

2) The Conclusion section should be re-written.   The majority of this section belongs in either the methods or results section and should not be in the conclusions section.

Author Response

Thanks to the reviewer for his work and

Point 1  

A:You mention in line 253 that "The suitable and ideal material for chest wall prosthetic reconstruction should be rigid enough to prevent paradoxical movements, quite malleable, strong resistance to infection, biologically inert, radiolucent, and cheap".   However, your institutional preference (line 307) includes the use of biologic mesh and titanium bars (when needed).   These materials, while they may fulfill some of your criteria, are also significantly more expensive that polypropylene mesh and/or methylmethacrylate.

R: In line 253 we reported what should be the ideal characteristics of a material for the reconstruction of the chest wall. However, all these characteristics are impossible to find in a single material. Biological meshes and titanium bar are more expensive than other materials but in chest wall tumors  we prefer to use these for excellent results both in the short and medium term. However we also use methylmethacrylate mesh. especially for the repair of lung hernias.

  1. A: Do you recommend biologic materials in ALL cases of chest wall reconstructions for defects >5cm, +3/4 ribs?   Are there situations where you still utilize synthetic materials?
  2. R: We recommend the use of biological materials in all types of reconstruction. Biological meshes can also be used for extensive thoracic and sternal reconstructions. In extensive reconstruction we are used to use biological meshes together with titanium bars protected by muscle flap. It is important to reiterate that to date there are no specific guidelines and the type of prosthetic material used often depends on the experience and preferences of the surgeon.

  1. A: You also mention the higher rates of infections in synthetic meshes.   Do you have institutional data that shows a significantly lower infection rate that can be added to this paper to support your use of these materials in all of your chest wall reconstructions?
  2. R: In our experience we have had several cases of infection of synthetic prosthetic materials, but we have no data available.However not even a case of infection after implantation of biological prostheses

Point 2

 A: The Conclusion section should be re-written.   The majority of this section belongs in either the methods or results section and should not be in the conclusions section.

 R: In the conclusions section we tried to summarize the most important aspects on the resection and reconstruction of the chest wall tumors to introduce 3D printing technology, innovative reconstruction technique

Round 2

Reviewer 1 Report

Dear Authors,

Thanks for your reply to my comment. Given that this is a state of art article, I now agree with the structure of manuscript and find the modified title more suitable.

I would recommend this manuscript for publication.

Your Sincerely